

# Epistemic uncertainties and natural hazard risk assessment. 2. What should constitute good practice?

Keith J. Beven[1,2], Willy P. Aspinall[3], Paul D. Bates[4], Edoardo Borgomeo[5], Katsu Goda[7], Jim W. Hall[5], Trevor Page[1], Jeremy C. Phillips[3], Michael Simpson[8], Paul J. Smith[1,6], Thorsten Wagener[7,9], and Matt Watson[3]

[1] Lancaster Environment Centre, Lancaster University, Lancaster, UK

[2] Department of Earth Sciences, Uppsala University, Uppsala, Sweden

[3] School of Earth Sciences, Bristol University, Bristol, UK

[4] School of Geographical Sciences, Bristol University, Bristol, UK

[5] Environmental Change Institute, Oxford University, UK

[6] European Centre for Medium-Range Weather Forecasting, Reading, UK

[7] Department of Civil Engineering, Bristol University, Bristol, UK

[8] Department of Mathematics and Computer Science, Exeter University, Exeter, UK

[9] Cabot Institute, University of Bristol, UK

*Correspondence to*: Keith J. Beven (k.beven@lancaster.ac.uk)

**Abstract.** Part 1 of this paper has discussed the uncertainties arising from gaps in knowledge or limited understanding of the processes involved in different natural hazard areas. Such deficits may include uncertainties about frequencies, process representations, parameters, present and future boundary conditions, consequences and impacts, and the meaning of observations in evaluating simulation models. These are the epistemic uncertainties that can be difficult to constrain, especially in terms of event or scenario probabilities, even as elicited probabilities rationalised on the basis of expert judgements. This paper reviews the issues raised by trying to quantify the effects of epistemic uncertainties. Such scientific uncertainties might have significant influence on decisions made, say, for risk management, so it is important to examine the sensitivity of such decisions to different feasible sets of assumptions, to communicate the meaning of associated uncertainty estimates and to provide an audit trail for the analysis. A conceptual framework for good practice in dealing with epistemic uncertainties is outlined and implications of applying the principles to natural hazard science are discussed.

## 1. Introduction

Part 2 of this paper constitutes a discussion of some of the issues raised by the review of different natural hazard areas in Part 1, with a view to addressing the question of what should constitute good practice in dealing with knowledge-related uncertainties in natural hazards assessment. For good epistemic reasons, there can be no definitive answer to the question,





only a variety of views. Thus what follows should be considered as an opinion piece, discussing some of the precepts held by the authors or opinions expressed by others elsewhere, any of which could be subject to revision in the future. We would argue, however, that an open discussion of the issues is valuable in itself at this stage.

There are, perhaps, a number of things on which we can agree:

- That epistemic uncertainties are those that are not well determined by historical observations, including where those observations are used in the formulation and conditioning of predictive models or simulators.
- That they arise in all stages of risk analysis which, in Part 1 of this paper and elsewhere (Rougier et al., 2013), have been characterised in terms of assessing Hazard, Footprint and Loss.
- That they might not have simple statistical structure and may not be bounded, complete or exhaustive, leaving open the potential for future surprise.
- That any analysis will be conditional on the assumptions about the hazard model, the footprint model and the loss model components AND about the nature of the errors associated with such model components.

Clearly, at each analysis stage, there is the potential for different methodological choices by the analyst and, since these decisions are necessarily not well determined by historical data (and there may be an expectation of future change or surprise), there can be no single right answer. Thus an evaluation of the potential impacts of feasible sets of assumptions, and the communication of the meaning of the resulting uncertainties to users of the analysis, should be important components of good practice. In what follows we discuss a conceptual framework for dealing with epistemic uncertainty in 20 natural hazards risk assessments, and express some opinions on what might constitute good practice in framing an analysis and the communication of the results to users.

**2. A framework for good practice in natural hazards modelling**

We suggest that a general framework for natural hazards modelling should include the following steps:

1. Establishing the purpose and framing the nature of the risk analysis to be undertaken in terms of the hazard, footprint, and loss to be considered.
2. Evaluating the information content of the available data.
3. Eliciting opinions about sources of uncertainty and the potential for future surprise.
4. Choosing a methodology and defining workflows for applying the method correctly and accurately.
5. Assessing whether a decision is robust to the chosen assumptions through some form of sensitivity analysis.
6. Communicating the meaning of the uncertainty analysis through a condition tree and audit trail, and visualising the outcomes for effective decision-making.





The following discussion is structured following these six steps.

## 2.1 Framing the nature of the risk analysis

The first, essential step when framing a risk analysis in natural hazards is to establish a defined purpose for the analysis.
This is usually resolved by discussion with the 'problem owner' or stakeholder group who will make use of the findings, and therefore their requirements should determine, in the main, what hazards, magnitudes, footprints and losses are to be considered. In most cases, this also involves bounding the problem, i.e. differentiating between what is to be included and what is to be considered beyond the analysis. Often, these latter decisions will be based on expert judgement, either informally or as a formal elicitation exercise. Historically, the framing of many risk analyses is already institutionalised as a
set of rules or statutory requirements. In several countries there are defined workflows for different types of risk, with some similarities of approach but differences in detail. Framing the problem has become more difficult with a greater recognition that the risk might not be stationary into the future. Future change is one of the most important sources of epistemic uncertainty in many natural hazards, especially for floods, droughts, landslides, and wind storms..

When analysing natural hazards, we are always interested in more extreme events, and the analysis of the risk of such events for use in decision making is made much simpler by considering that different (uncertain) components of the risk can be defined in terms of probability distributions. This is effectively allowing that all epistemic uncertainties that are recognised can be treated analogously to aleatory uncertainties. This is usually the form of output of an expert elicitation exercise, for example (Section 2.3 below). Such an approach allows for the full power of statistical theory to be applied, since the
analysis is based on the manipulation of probabilities, but as a result, the risk assessment depends heavily on the distributional assumptions that are made, notably in the tail behaviour of the extremes (of both magnitudes and potential losses). Since samples of extreme events are, by their very nature, generally small then there has to be some epistemic uncertainty about the appropriate distributions to be applied, even if some countries have institutionalised certain distributions as a convenient standard practice in this respect, at least for magnitudes (see Section 3 below). Footprints and
losses are often treated more deterministically (as discussed, for example, in Rougier et al., 2013).

Natural hazard risk assessments have muddled along for many decades on this basis without undue criticism, except from communities that get impacted because little or no protection is provided, or because any protection employed is subsequently deemed to have been excessive. In such situations, the assessment, and the analyst, can always be defended by
invoking the capricious nature of reality. This is because the magnitude of a particular event is usually evaluated in terms of its frequency of occurrence (as a "return period" or "annual exceedance probability"); then, if a new event comes along that is bigger than that estimated as the reference event for a risk analysis, it has, by definition, a lower probability of exceedance. It may be a surprise if two events of low probability occur in quick succession, but even then there is always a small but





finite statistical probability of that occurring under the assumptions of the analysis. Effectively, the analyst cannot be wrong. Post-hoc, of course, the new event(s) can also be used to revise the risk analysis. This has, perhaps, been one reason why there has been little pressure to treat epistemic uncertainties more explicitly when, again for good epistemic reasons, it is difficult to know exactly what assumptions to make.

## 2.2 Evaluating the information content of the available data.

The assessment of the potential for future natural hazard events frequently involves the combination of model outputs with data from historical events. Often, only the simplest form of frequency analysis is used where the model or simulator is a chosen distribution function for the type of event being considered, and where the data are taken directly from the historical

10 record. Such data are often used uncritically (since they are generally the only information available to condition local uncertainty estimates) but it is important to recognise that both model and data will be subject to forms of epistemic uncertainty. In some cases this might lead to the situation that either the model or the data might be 'disinformative' in assessing the future hazard. A frequency distribution that underestimates the heaviness of the upper tail of extreme events, for example, might lead to underperformance of any protection measures or underestimation of the zone at risk.

Similarly any data used to condition the risk might not always be sufficiently certain to add real information to the assessment process. In the context of conditioning rainfall-runoff simulator parameters, for example, Beven et al. (2011) and Beven and Smith (2015) have demonstrated how some event data suggest that there is more estimated output from a catchment area in northern England than the inputs from many events recorded in three rain gauges within that catchment.

20 No simulator that maintains mass balance (as is the case with most rainfall-runoff simulators) will be able to predict more output than input, so that including those events in conditioning the simulator would lead to incorrect inference (at least for catchments with no significant groundwater storage). Why the data do not satisfy mass balance could be because the rain gauges underestimate the total inputs to the catchment, or that the discharge rating curve (which relates measured water levels to discharge at an observation point) overestimates the discharges when extrapolated to larger events.

Other examples come from the uncertain identification of inundated areas by either post-event surveys or remote sensing (e.g. Mason et al., 2007), and observational data on the evolution of tropical cyclones (e.g. Hamill et al., 2011). These are just some examples of the more general problem of assessing the information content of data that is subject to epistemic uncertainties. Normal statistical methods of assessing information content (e.g. entropy measures) may not be appropriate

30 when the error characteristics have complex structure, are non-stationary in time or space, or are physically inconsistent (Beven and Smith, 2015).



### 2.3 Eliciting opinions about sources of uncertainty and the potential for future surprise.

Expert judgement is one of the main ways of trying to take account of epistemic uncertainties in natural hazards risk assessment. Experts can be asked to assign estimates of the probabilities or possibilities of potential outcomes. Such estimates will be necessarily judgement-based and, while they may be epistemically incomplete in some regards, good
experts will base their judgements on a comprehensive synthesis of available evidence. However, when epistemic uncertainties dominate in natural hazards assessment, some individual experts will sometimes be surprised by events or outcomes and, in consequence, their judgements then turn out to be quite inaccurate or uninformative. To mitigate such situations it is therefore important to include the judgements of as many experts as possible. This, in turn, can raise questions about the independence of experts who have similar backgrounds and training, and about whether more weight
should be given to the judgements of some experts relative to others, and about how this should be done. For instance, different levels of experience may be an important factor when it comes to expert judgement performance.

Cooke (2014) gives a good recent summary of some of the issues involved in expert elicitation (see, in particular, the extensive supplementary material to that paper). He points out that both Knight (1921) and Keynes (1921) suggested that the
use of elicited expert probabilities might be a working practical solution to dealing with these types of "real" uncertainties. A variety of methods have been proposed for assessing the value of experts, and combining their judgements in an overall risk assessment (see Cooke, 1991; O'Hagan et al., 2006; Aspinall and Cooke, 2013). Cooke (2014) includes a review of post-elicitation analyses that have been carried out seeking to validate assessments conducted with the Classical Model Structured Expert Judgment (SEJ) method (Cooke, 1991). This appraisal of applications in a variety of fields includes some
for natural hazards (see Cooke and Goossens, 2008; Aspinall and Cooke, 2013; Aspinall and Blong, 2015).

A recent application of the Classical Model SEJ has provided an unprecedented opportunity to test the approach, albeit in a different field. The World Health Organization (WHO) undertook a study involving 72 experts distributed over 134 expert panels, with each panel assessing between 10 and 15 calibration variables concerned with foodborne health hazards (source
attribution, pathways and health risks of foodborne illnesses). Calibration variables drawn from the experts' fields were used to gauge performance and to enable performance-based scoring combinations of their judgments on the target items. The statistical accuracy of the experts overall was substantially lower than is typical with a Classical Model SEJ, a fact explained by operational limitations in the WHO global elicitation process. However, based on these statistical accuracy and informativeness measures on the calibration variables, performance-based weighted combinations were formed for each
panel. In this case, in-sample performance of the performance-based combination of experts (the 'Performance Weights Decision Maker' PW DM) is somewhat degraded relative to other Classical Model SEJ studies (e.g. Cooke and Coulson, 2015) but, this said, performance weighting still out-performed equal weighting ('Equal Weights Decision Maker' EW DM) (Aspinall et al., 2016).



Because a large number of experts assessed similar variables it was possible to compare statistical accuracy and informativeness on a larger dataset than hitherto (Aspinall et al., 2016). For certain foodborne health hazards, some regions of the world were considered interchangeable, and so a panel could be used multiple times. Also, many experts participated

in several distinct panels. For these reasons, any statistical analysis of results that considers the panels as independent experiments is impossible, and so proper out-of-sample analysis was infeasible.

However, this extensive study has provided new perspectives on the efficacy of SEJ. Most significant in this data set was the negative rank correlation between informativeness and statistical accuracy, and the finding that this correlation weakens

when expert selection is restricted to those experts who are demonstrated by the Classical Model empirical calibration formulation to be more statistically accurate (Aspinall et al., 2016). These findings should motivate the development and deployment of enhanced elicitor and expert training, and advanced tools for remote elicitation of multiple, internationally-dispersed panels – demand for which is growing in many disciplines (e.g. low probability high consequence natural hazards; climate change impacts; carbon capture and storage risks).

Such expert elicitation exercises are one way of deciding on an appropriate methodology and sets of assumptions for an analysis, with their associated probabilities, which will define the workflow for the analysis (see next Section). Such probabilities, as noted, can be difficult to estimate and will necessarily involve some subjectivity. As such, they are more analogous to gambling odds than empirical probabilities (even if empirically they have been assessed over multiple experts),

and might best be treated as such. These odds can follow the axioms of probability - the original formulation of Bayes theorem was expressed in terms of defining the odds for a given proposition or hypothesis (Bayes, 1763) - but are less likely to be considered as complete for the convenience of analysis (as is the case for most fitted probability distributions). Thus some element of the potential for surprise can be retained.

**2.4   Choosing a methodology and defining workflows for applying a method correctly**

The review of Part 1 of this paper shows that there exist some differences in practice between different hazard areas, in part dependent on the availability of data and the potential for doing useful forecasting as well as simulation. It is helpful to distinguish three types of uncertainty analysis (e.g. Beven, 2009).


The first is a forward analysis where the outputs depend entirely on propagating prior assumptions about the sources of uncertainty, albeit that those prior assumptions might be derived from historical sequences. Risk assessments of droughts, dam safety, landslides, ground motion from earthquakes, and tsunamis tend to be of this type. When trying to draw


inferences about future hazard occurrences it can be difficult to define those prior assumptions so that, in such an analysis, the decisions about how far those assumptions truly reflect potential sources of epistemic uncertainty (for example in climate change projections) are paramount. This is, necessarily, an exercise in expert judgement, which may be formalised in the type of expert elicitation exercise discussed earlier.

The second form of analysis involves conditioning the prior estimates of uncertainty for a simulation model on observational data. Flood inundation maps (using historical flood outlines and discharge estimates), and the inversion methods used for identifying earthquake ruptures, and source terms for ash cloud simulations are of this type. In general, such methods will help to constrain model uncertainties, but will be dependent on both the range of models considered and the way in which

they are evaluated relative to the available observations. A number of conditioning methodologies are available including formal Bayes methods (Bernado and Smith, 2009); Bayes linear methods (Goldstein and Wooff, 2007); Approximate Bayesian Computation (ABC, Vrugt and Sadegh, 2013; Nott et al., 2014) and Generalised Likelihood Uncertainty Estimation (GLUE, Beven and Binley, 1992, 2014; Blazkova and Beven, 2009). The latter can make use of formal and informal likelihood measures and limits of acceptability, as well as alternatives to Bayes rule in combining different model

evaluations.

Because epistemic uncertainties are involved, including the potential for non-stationary bias and error characteristics and unknown commensurability errors between observed and predicted variables, it might not always be good practice to use formal statistical likelihood measures in such evaluations. Indeed, epistemic uncertainties make it difficult to test such

models as hypotheses in rigorous ways, and may mean that multiple different model structures might all be consistent with the available observations (e.g. Beven, 2002, 2006, 2012). There may also be issues of whether the available models are fit-for-purpose when compared with observations, even when the uncertainty associated with those observations is taken into account. We should certainly be wary of using overly simple statistical analyses without testing the validity of the assumptions made, or of disregarding important sources of uncertainty (such as is often done in flood risk analysis for

example, where uncertainty in historical flood magnitudes is generally neglected). There will also be aspects of an analysis of epistemic uncertainties that may not be amenable to such conditioning, for example assumptions about potential future climate scenarios or other future boundary conditions. These will depend on the type of expert judgment or elicitation in a way similar to a forward uncertainty analysis.

The third form of uncertainty analysis can be used when the interest is in forecasting a hazard into the near future and when observables are available in real time to allow the use of data assimilation to constrain prediction uncertainties. A variety of data assimilation methods is available, from the variational methods commonly used in weather forecasting, to ensemble Kalman filters and Particle filters. Such methods are used in real time forecasting of floods, ash clouds and wind storms (see Part 1 of this paper). It is perhaps instructive in the context of a discussion of epistemic uncertainties that in generating an



ensemble of future weather forecasts, singular vector techniques are used to choose more extreme perturbations in formulating the members of the ensemble, so as to stand a greater chance of bracketing the potential range of future weather over the lead time of a few days. The ensemble members should therefore be considered to be of unknown probability, even if the outputs are sometimes interpreted in probabilistic ways (such as in decisions about alert status in the European Flood

Awareness System based on simulated river flows forced by ECMWF ensemble predictions of rainfalls when compared to a long term historical reanalysis, see Pappenberger et al., 2013).

All of these forms of uncertainty analysis strategies and workflows are subject to epistemic uncertainties. The first form of forward analysis depends wholly on how they are incorporated into prior assumptions, the second allows for some

conditioning of outcomes on available data, but cannot easily allow for differences in the future, the third can probably allow best for mis-specified assumptions because of the potential for continuous updating and a limited time frame for the forecasts. However, in all cases it will be good practice to assess the sensitivity of the outcomes to a range of feasible assumptions (e.g. Saltelli, 2002; Tang et al., 2007; Saltelli et al., 2008; Pianosi et al., 2015; Savage et al., 2016).

**2.5   Assessing whether a decision is robust to the chosen assumptions.**

The primary reason for making uncertainty assessments for evaluating risk in natural hazards is because taking account of uncertainty might make a difference to the decision that is made (e.g. Hall and Solomatine, 2008; Rougier and Beven, 2013; Hall, 2003; Simpson et al., 2016). For many decisions a complete, thoughtful, uncertainty assessment of risk might not be justified by the cost in time and effort. In other cases, the marginal costs of such an analysis will be small relative to the

potential costs and losses, so a more complete analysis using more sophisticated methods would be justifiable, including using expert elicitations in defining the assumptions of the relevant workflow.

Formal risk-based decision making requires probabilistic representations of both the hazard and consequence components of risk, i.e. an assumption that both hazard and consequences can be treated as aleatory variables, even if the estimates of the

probabilities might be conditional and derived solely from expert elicitation or estimates of odds. The difficulty of specifying odds or probabilities for epistemic uncertainties means that any resulting decisions will necessarily be conditional on the assumptions (as discussed, for example, by Pappenberger and Beven, 2006; Beven, 2009; Sutherland et al., 2013; Rougier and Beven, 2013; and Juston et al. 2013).

It also leaves scope, however, for other methodologies for uncertainty assessment, including fuzzy possibilistic reasoning, Dempster-Shafer evidence theory, Prospect Theory and Info-gap methods (see Shafer, 1976; Kahneman and Tversky, 1979; Halpern, 2003; Hall, 2003; Ben-Haim, 2006; Wakker, 2010). There is some overlap between these methods, for example





Dempster-Shafer evidence theory contains elements of fuzzy reasoning and imprecise reasoning, while both Prospect Theory and Info-Gap methods aim to show why non-optimal solutions might be more robust to epistemic uncertainties than classical risk based optimal decision making. There have been just a few applications of these methods in the area of natural hazards, for example: Prospect Theory to seismic design optimisation (Goda and Hong, 2008); and Info-Gap theory to flood defence

assessments (Hine and Hall, 2010), drought assessments in water resource management (Korteling et al., 2013), and earthquake resistant design criteria (Takewaki and Ben-Haim, 2005). All such methods require assumptions about the uncertainties to be considered, so can be usefully combined with expert elicitation.

By definition we cannot distinguish between different choices about how to represent epistemic uncertainty through

comparison with observations. However, we can always test how much it matters if we make different assumptions, if we change boundary conditions, or if we include the potential for data to be wrong. While most global sensitivity analysis approaches assume that we can define some probability distribution to characterize the potential variability of the inputs into the analysis, we might still gain useful information concerning whether the epistemic uncertainty related to an individual input might even matter from such a sensitivity analysis (see, for example, the suggestion of using conditional risk

exceedance probability curves in Rougier and Beven, 2013). We now have formal approaches to sensitivity analysis that enable us to include discrete choices, e.g. different probability distribution functions, imprecise probabilities or process representations (e.g. Hall, 2006; Baroni and Tarantola, 2014; Savage et al., 2016) or that allow us to explore the impact of distributions that are practically unconstrained due to a lack of observations (e.g. Prudhomme et al., 2010; Singh et al., 2014; Almeida et al., 2016). Alternatively, we can directly select an approach that attempts to find robust decisions in the presence

of poorly bounded uncertainties (e.g. Bryant and Lempert, 2010; Steinschneider et al., 2015).

However, while there are important advantages in treating a risk analysis or reliability problem as a problem in assessing probabilities, this might lead to a degree of confidence in the outcomes that may not be justified. That will especially be the case when a statistical analysis is too simplified, for example in fitting magnitude-frequency distributions to historical data without allowing for the uncertainty in the data, the uncertainty in tail behaviour, the possibility of multiple underlying

drivers in the hazard, or the potential for joint hazard occurrences. Asymptotic extreme value distributions (such as the Gumbel distribution which strictly applies only to independent extreme values) are often applied to small samples without consideration of the potential for non-stationarity and dependencies in occurrences, or the resulting uncertainty in the frequency estimates of extremes.

Similar issues arise in the evaluation of model simulations against historical data. It is often assumed that the residuals have a simple statistical structure (e.g. independent and Gauss distributed with constant variance), leading to a simple formal likelihood function. Where these assumptions are not valid, overconfidence in the inference will result (e.g. Beven and





Smith, 2015; Beven, 2016). These are (common) examples of poor practice, especially where the assumptions are not checked for validity.

This should not, however, be considered as an argument against the use of probabilistic methods in natural hazards risk
analysis, just that better statistical training and more sophisticated analyses might be required to produce more robust inference. Nearing et al. (2016), in a hydrological context, have argued that probabilities are the only consistent framework for uncertainty assessment. The first stage in this is to check the validity of the assumptions, wherever that is possible. Where model errors can be evaluated against historical data, they should be checked for consistency, stationarity, and residual structure. Where there is a cascade of nonlinear model components, the effect of each component on the structure of
the output errors should be assessed (Gaussian input errors will not remain Gaussian when processed through a nonlinear model structure, and the resulting non-stationarity in error characteristics might be difficult to represent in any likelihood evaluation or conditioning exercise). In such cases a more formal, sophisticated, analysis will be justified with at least a sensitivity analysis of the outcomes to different assumptions when assessing robustness.

It is clear from discussions amongst the authors and in the refereeing of this paper, that opinions vary as to whether a more sophisticated statistical analysis is already sufficient. It is certainly more difficult to assess the impact of analysing the probabilities as if they were incomplete. This leaves open the potential for future surprise when the epistemic uncertainties are such that the probabilities are not well defined. Falling outside historical data support, a surprise event is intrinsically non-probabilistic (and might not be easily be foreseen or even perceived by expert elicitation. This is epistemic uncertainty
as a form of "unknown unknowns" (or very hard to evaluate unknowns). The potential for some form of surprise event is often envisaged, however, which suggests that where this might have catastrophic consequences it should be made an explicit part of a robust analysis.

Since surprise does not enter easily into a risk-based decision making framework, the question is how to allow for
incompleteness? Techniques might include an extension of the expert elicitation approach to consider the unexpected; a form of Info-Gap analysis, looking at impact of more extreme events and the opportuneness benefit if they do not occur (Ben-Haim, 2006); an evaluation of the most extreme event to be expected, even if of unknown probability; or a type of factor of safety approach based on costs of being precautionary. All of these approaches are associated with their own epistemic uncertainties, but addressing the question of surprise and whether the probabilities of a risk analysis are properly
defined or incomplete is, at least, a step in the direction of good practice.

Potential climate change impacts on future risk represent one example where the problem of incomplete uncertainties and discrete choices is central to the analysis. Here, we can make assumptions and give different projections (from different Global Circulation Models, or downscaling procedures, or ways of implementing future change factors) equal probability.



But this is a case where the range of possibilities considered may not be complete (e.g. Collins et al., 2012) and climate change might not be the only factor affecting future risk (Wilby and Dessai, 2010). We could invoke an expert elicitation to say whether a particular projection is more likely than another and to consider the potential for changes outside the range considered, but, as noted earlier, it can be difficult sometimes to find experts whose views are independent of the various

modelling groups. It is also not clear whether the modelling groups themselves fully understand the potential for uncertainties in their simulations, given the continuing computing constraints on this type of simulation.

Consideration of climate change risks in these contexts has to confront a trio of new quantitative hazard and risk assessment challenges: micro-correlations, fat tails and tail dependence (e.g. Kousky and Cooke, 2009). These are distinct aspects of

loss distributions which challenge traditional approaches to managing risk. Micro-correlations are tiny or local correlations that, as individual factors, may be harmless, but very dangerous when they coincide and operate in concert to create extreme cases. Fat tails can apply to losses whose probability declines slowly, relative to their severity. Tail dependence is the propensity of a number of extreme events or severe losses to happen together. If one does not know how to detect these phenomena or relationships, it is easy to not see them, let alone cope with or predict them adequately. Dependence

modelling is an active research topic, and methods for dependence elicitation are still very much under development (e.g. Morales et al., 2008).

It is hard to believe that current natural hazard and climate models are not subject to these types of epistemic uncertainties. In such circumstances, shortage of empirical data inevitably requires input from expert judgment to determine relevant

scenarios to be explored. How these behaviours and uncertainties are best elicited can be critical to a decision process, as differences in efficacy and robustness of the elicitation methods can be substantial. When performed rigorously, expert elicitation and pooling of experts' opinions can be powerful means for obtaining rational estimates of uncertainty.

### 2.6 Communicating the meaning of uncertainty analysis and visualising the outcomes for effective decision-making.

Given all the assumptions that are required to deal with epistemic uncertainties in natural hazard and risk analysis, there are real issues about communicating the meaning of an uncertainty assessment to potential users (e.g. Faulkner et al., 2007, 2014). Good practice in dealing with different sources of uncertainty should at least involve a clear and explicit statement of the assumptions of a particular analysis (Hall and Solomatine, 2008). Beven and Alcock (2012) suggest that this might be expressed in the form of condition trees that can be explicitly associated with any methodological workflow. The condition

tree is a summary of the assumptions and auxiliary conditions for an analysis of uncertainty. The tree may be branched in that some steps in the analysis might have subsidiary assumptions for different cases. The approach has two rather nice features. Firstly, it provides a framework for the discussion and agreement of assumptions with experts and potential users of the outcomes of the analysis. This then facilitates communication of the meaning of the resulting uncertainty estimates to





those users.   Secondly, it provides a clear audit trail for the analysis that can be reviewed and evaluated by others at a later date.

The existence of the audit trail might focus attention on appropriate justification for some of the more difficult assumptions

that need to be made; such as how to condition simulator outputs using data subject to epistemic uncertainties and how to deal with the potential for future surprise (see Beven and Alcock, 2012).   Application of the audit trail in the forensic examination of extreme events as and when they occur might also lead to a revision of the assumptions as part of an adaptive learning process for what should constitute good practice.

Such condition trees can be viewed as parallel to the logic trees or belief networks used in some natural hazards assessments, but focussed on the nature of the assumptions about uncertainty that leads to conditionality of the outputs of such analyses. Beven et al. (2014) and Beven and Lamb (2017) give examples of the application of this methodology to mapping inundation footprints for flood risk assessment.

In understanding the meaning of uncertainty estimates, particularly when epistemic uncertainties are involved, understanding the assumptions on which the analysis is based is only a starting point.   In many natural hazards assessments those uncertainties will have spatial or space-time variations that users need to appreciate.   Thus visualisation of the outcomes of an uncertainty assessment has become increasingly important as the tools and computational resource available have improved in the last decade and a variety of techniques have been explored to represent uncertain data (e.g. Johnson and

Sanderson, 2003; MacEachren et al., 2005; Pang, 2008; Kunz et al., 2011; Friedemann et al., 2011; Spiegelhalter et al., 2011; Spiegelhalter and Reisch, 2011; Jupp et al., 2012; Potter et al., 2012).

One of the issues that arise in visualisation is the uncertainty induced by the visualisation method itself, particularly where interpolation of point predictions might be required in space and/or time (e.g. Agumya and Hunter, 2002; Couclelis, 2003).

The interpolation method will affect the user response to the visualisation in epistemically uncertainty ways.   Such an effect might be small, but it has been argued that, now that it is possible to produce convincing virtual realities that can mimic reality to an apparently high degree of precision, we should be wary about making the visualisations too good so as not to induce an undue belief in the simulator predictions and assessments of uncertainty in the potential user (e.g. Dottori et al., 2013; Faulkner et al., 2014).


Some examples of visualisations of uncertainty in natural hazard assessments have been made for flood inundation (e.g. Beven et al., 2014; Faulkner et al., 2014, Leedal et al., 2010; Pappenberger et al., 2013); seismic risk (Bostrom et al., 2008); tsunami hazard and risk (Goda and Song, 2016); volcanic hazard (Marzocchi et al., 2010; Wadge and Aspinall, 2014; Baxter et al., 2014); and ice-sheet melting due to global temperature change (Bamber and Aspinall 2013).   These are all cases where



different sources of uncertainty have been represented as probabilities and propagated through a model, a simulator or cascade of simulators to produce an output (or outputs) with enumerated uncertainty. The presentation of this uncertainty can be made in different ways and can involve interaction with the user as a way of communicating meaning (e.g. Faulkner et al., 2014). But, as noted in the earlier discussion, this is not necessarily an adequate way of representing the "deeper"

epistemic uncertainties, which are not easily presented as visualisations (Spiegelhalter et al., 2011).

### 3. Epistemic Uncertainty and Institutionalised Risk Assessments

The framework for good practice discussed in the previous section allows for considerable freedom in the choice of

methodology and assumptions, but it has been suggested that these choices must be justified and made explicit in the workflow, condition tree, and audit trail associated with an analysis. This contrasts with an approach that has been common in many fields of natural hazards assessment where those choices have been institutionalised, with specified rules usually based on expert judgement, to provide science-informed planning for dealing with potentially catastrophic natural hazards.

One example of such an institutionalised approach is in the use of estimates of annual exceedance probabilities that are used for natural hazards planning in different countries. In the UK, frequency-based flood magnitude estimates are used for flood defence design and planning purposes in rather deterministic ways. For fluvial flooding defences are designed to deal with the rare event (annual exceedance probability of less than 0.01). The footprint of such an event is used to define a planning zone. The footprint of a very rare event (annual exceedance probability of less than 0.001) is used to define an outer zone.

Other countries have their own design standards and levels of protection.

While there is no doubt that both deterministic and frequency assessments are subject to many sources of epistemic uncertainty, such rules can be considered as structured ways of dealing with those uncertainties. The institutionalised, and, in some cases, statutory, levels of protection are then a political compromise between costs and perceived benefits. Such an

approach is well developed in earthquake engineering in assessing life cycle costs and benefits, at least for uncertainties that can be defined as probability distributions (e.g. Takahashi et al., 2004). Flood defence is also an example where the analysis can be extended to a life cycle risk-based decision analysis, with costs and benefits integrated over the expected frequency distribution of events (Sayers et al., 2002; Voortman et al., 2002; Hall and Solomatine, 2008). In the Netherlands, for example, where more is at risk, fluvial flood defences are designed to deal with an event with an annual exceedance

probability of 0.0008, and coastal defences to 0.00025. In doing so, of course, there is an implicit assumption of stationarity of the hazard when based only on the analysis of current and historical data.

There have also been practical approaches suggested based on estimating or characterising the largest event to be expected in any location of interest. Such a maximal event might be a good approximation to very rare events, especially when the



choice of distribution for the extremes is bounded. In hydrological applications the concepts of the probable maximum precipitation and probable maximum flood have a long history (e.g. Hershfield, 1963; Newton, 1983; Hansen, 1987; Douglas and Barros, 2003; Kunkel et al., 2013) and continue to be used, for example in dam safety assessments (e.g. Graham, 2000, and Part 1 of this paper). In evaluating seismic safety of critical infrastructures (e.g. nuclear power plants and dams), there

have been some who would move away from probabilistic assessment of, say, future earthquake magnitudes, preferring the concept of a deterministic maximum estimate of magnitude (McGuire, 2001; Panza et al., 2008; Zucollo et al., 2011). These "worst case" scenarios can be used in decision making but are clearly associated with their own epistemic uncertainties and have been criticised because of the assumptions that are made in such analyses (e.g. Koutsoyiannis, 1999; Abbs, 1999; Bommer, 2002).

An important problem with this type of institutionalised analysis is that sensitivity to the specified assumptions is rarely investigated and uncertainties in such assessments are often ignored. A retrospective evaluation of the recent very large Tōhoku, Japan, earthquake indicates that the uncertainty of the maximum magnitude in subduction zones is considerable and, in particular, it is argued that the upper limit in this case should have been considered unbounded (Kagan and Jackson,

2013). This reasoning, however, has the benefit of hindsight.   Earlier engineering decisions relating to seismic risk at facilities along the coast opposite the Tōhoku subduction zone had been made on the basis of work by Ruff and Kanamori (1980), repeated by Stern (2002). Stern reviewed previous studies and described the NE Japan subduction zone (Figure 7 of Stern, 2002) as a "good example of a cold subduction zone", denoting it the "old and cold" end-member of his thermal models. Relying on Ruff and Kanamori (1980), Stern re-presented results of a regression linking 'maximum magnitude' to

subduction zone convergence rate and age of oceanic crust. This relationship was said to have a "strong influence … on seismicity" (Stern, 2002), and indicated a modest maximum moment magnitude (Mw) of 8.2 for the NE Japan subduction zone. It is not surprising, therefore, that such authoritative scientific sources were trusted for engineering risk decisions.

This said, MacCaffrey (2008) had pointed out, before the Tōhoku earthquake, that the history of observations at subduction

zones is much shorter than the recurrence times of very large earthquakes, suggesting the possibility that any subduction zone may produce earthquakes larger than Mw 9.   In respect of the Ruff and Kanamori relationship, epistemic uncertainties (as discussed in Section 2.5 above) were unquestionably present, and almost certainly large enough, to undermine the robustness of that regression for characterising a long-term geophysical process.   Thus, epistemic uncertainties for any purported maximum event should be carefully discussed from both probabilistic and deterministic viewpoints, as the

potential consequences due to gross underestimation of such events can be catastrophic (e.g. Goda and Abilova, 2016).

These deterministic maximal event approaches do not have probabilities associated with them, but can serve a risk averse, institutionalised role in building design or the design of dam spillways, say, without making any explicit uncertainty estimates. In both of these deterministic and probabilistic scenario approaches, the choice of an established design standard



is intended to make some allowance for what is not really known very well, but with the expectation that, despite epistemic uncertainties, protection levels will be exceeded sufficiently rarely for the risk to be acceptable.

Another risk averse strategy for coping with lack of knowledge is in the factors of safety that are present in different designs for protection against different types of natural hazard, for example in building on potential landslide sites when the effective parameters of slope failure simulators are subject to significant uncertainty.   In flood defence design, the concept of "freeboard" is used to raise flood embankments or other types of defences.  Various physical arguments can be used to justify the level of freeboard (see, for example, Kirby and Ash, 2000) but the concept also serves as a way of institutionalising the impacts of epistemic uncertainty.

Such an approach might be considered reasonable where the costs of a more complete analysis cannot be justified, but this can also lead to overconfidence in cases where the consequences of failure might be severe.  It such circumstances, and for managing risks, it will be instructive to make a more detailed analysis of plausible future events, their impacts and consequences.

## 4.  Epistemic uncertainty and doing science

How to define different types of uncertainty, and the impact of different types of uncertainty on testing scientific models as hypotheses has been the subject of considerable philosophical discussion that cannot be explored in detail here (but see, for
example, Howson and Urbach, 1993; Mayo, 1996; Halpern, 2003; Mayo and Spanos, 2010; Gelman and Shalizi, 2013).  As noted earlier, for making some estimates, or at least prior estimates, of epistemic uncertainties we will often be dependent on eliciting the knowledge of experts.   Both in the Classical Model of Cooke (1991; 2014) and in a Bayesian framework (O'Hagan et al., 2006), we can attempt to give the expert elicitation some scientific rigour by providing empirical control on the how well the evaluation of the informativeness of experts has worked.  Empirical control is a basic requirement of any
scientific method and a *sine qua non* for any group decision process that aspires to be rational and to respect the axioms of probability theory.

Being scientific about testing the mathematical models that are used in risk assessments of natural hazards is perhaps less clear-cut. Models can be considered as hypotheses about the functioning of the real world system.   Hypothesis testing is
normally considered the domain of statistical theory (such as the severe testing in the error statistical approach of Mayo, 1996), but statistical theory (for the most part) depends on strongly aleatory assumptions about uncertainty that are not necessarily appropriate for representing the effects of epistemic sources of uncertainty (e.g. Beven and Smith, 2015; Beven, 2016).  Within the Bayesian paradigm, there are ways of avoiding the specification of a formal aleatory error model, such as in the use of expectations in Bayes linear methods (Goldstein and Wooff, 2007), in Approximate Bayesian Computation





(Diggle and Gratton, 1984; Vrugt and Sadegh, 2013; Nott et al., 2014), or in the informal likelihood measures of the GLUE methodology (Beven and Binley, 1992; 2014; Smith et al., 2008). It is still possible to empirically control the performance of any such methodology in simulating past data, but, given the epistemic nature of uncertainties this is no guarantee of good predictive performance in future projections. In particular, if we have determined that some events might be disinformative

for model calibration purposes, in forecasting or simulation we will not then know if the _next_ event would be classified as informative or disinformative if the observed data were made available, with important implications for prediction uncertainties (Beven and Smith, 2015).

In this context, it is interesting to consider what would constitute a severe test (in the sense of Mayo) for a natural hazard risk

assessment model. In the Popperian tradition, a severe test is one that we would expect a model could fail. However, all natural hazards models are approximations, and if tested in too much detail (severely) are almost certain to fail. We would hope that some models might still be informative in assessing risk, even if there are a number of celebrated examples of modelled risks being underestimated when evaluated in hindsight (see Part 1 of this paper). And, since the boundary condition data, process representations, and parameters characteristic of local conditions are themselves subject to epistemic

uncertainties, then any such test will need to reflect what might be feasible in model performance conditional on the data available to drive it and assess that performance. Recent sensitivity studies of Regional Climate Models, for example, have suggested that they cannot adequately reproduce high intensity convective rainstorms and consequently might not be fit-for-purpose in predicting conditions for future flooding induced by such events (e.g. Kendon et al., 2014; Meredith et al., 2015).

Some recent applications within the GLUE framework have used tests based on limits of acceptability determined from an assessment of data uncertainties before running the model (e.g. Liu et al., 2009; Blazkova and Beven, 2009). Such limits can be normalised across different types and magnitudes of evaluation variables. Perhaps unsurprisingly, it has been found that only rarely does any model run satisfy all the specified limits. This will be in part because there will be anomalies or disinformation in the input data (or evaluation observations) that might be difficult to assess a priori. This could be a reason

for relaxing the severity of the test such that only 95% of the limits need be satisfied (by analogy with statistical hypothesis testing) or relaxing the limits if we can justify not taking sufficient account of input error. In modelling river discharges, however, it has been found that the remaining 5% might be associated with the peak flood flows or drought flows which are the characteristics of most interest in natural hazards. Concluding that the model does not pass the limits test can be considered a good thing (in that we need to do better in finding a better model or improving the quality of the data, or

making a decision in a more precautionary way). It is one way of improving the science in a situation where epistemic uncertainties are significant.

This situation does not arise if there are no conditioning observations available so that only a forward uncertainty analysis is possible, but we should be aware in considering the assumptions that underlie the condition tree discussed earlier, that such a





forward model might later prove to be falsified by future observational data.  And, if we cannot argue away such failure, then it will be necessary to seek some other methodology for the risk assessment that  better accommodates gaps in our knowledge or understanding.

**5. Summary and Conclusions**

In assessing future risk due to natural hazards it is generally necessary to resort to the use of a model or simulator of some form, even if that is only a frequency distribution for expectations of future magnitudes of some hazard.  Even in that simple case, there will be limitations to the knowledge of what distribution should be assumed, especially when the database of past

events is sparse.  For risk-based decision-making the consequences of events must also be modelled in some way and these, equally, are liable to be subject to uncertainties due to limited knowledge.   Even though our simulation may be shown to match well a sample of past data, they may not perform adequately in future because of uncertainty about future boundary conditions and potential changes in system behaviour.   All of these (sometimes rather arbitrary) sources of epistemic uncertainty are inherently difficult to assess and, in particular, to represent as probabilities, even if we do recognise that those

probabilities might be judgement-based, conditional on current knowledge, and subject to future revision as expert knowledge increases.   As Morgan (1994) notes, throughout history decisions have always been made without certain knowledge, but mankind has mostly muddled through.

But, this rather underplays the catastrophic consequences of some poor decisions (including the recent examples of the

L'Aquila earthquake in Italy, and the Fukushima tsunami in Japan), so there is surely scope for better practice in future in trying to allow for: all models being intrinsically imprecise; most uncertainties being epistemic (even if some might be treated as if aleatory); all uncertainty estimates being conditional; a tendency for expert elicitations to underestimate potential uncertainties, and the potential for future surprises to happen.   Epistemic uncertainty suggests that we should be prepared for surprises and should be wary of both observational data and models which, while they might be the best we

have, might not be necessarily fit for purpose in some sense.  Surprises will occur when the probability estimates are incomplete, when the distribution tails associated with extremes are poorly estimated given the data available, or where subtle high dimensional relationships are not recognized or are ignored.

This then raises issues about the meaning of uncertainty estimates and how they might be interpreted by potential users,

stakeholders and decision makers (e.g. Sutherland et al., 2013).  Visualisations can be helpful in conveying the nature of uncertainty outputs but the deeper epistemic uncertainties might not be amenable to visualisations.   How to deal with epistemic uncertainties in all areas of natural hazards requires further research in trying to define good practice, particularly in assessing the models that are used in natural hazard risk assessments in a scientifically rigorous way.   Adopting the





condition tree, audit trail, rigorous sensitivity analysis and model hypothesis testing approaches suggested here would represent one beneficial step in that direction.

**Acknowledgements**

This work is a contribution to the CREDIBLE consortium funded by the UK Natural Environment Research Council (Grant NE/J017299/1). Thanks are due to Michael Goldstein, the referees and guest editor for comments on earlier versions of the paper.

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
