# Peer review of "Epistemic uncertainties and natural hazard risk assessment. 2. What should constitute good practice?"

_Natural Hazards and Earth System Sciences, 2017_

## Referee Comment (RC1) · M. Poole (Referee) · 12 Sep 2017

My name is Mike Poole and I am a Senior Research Manager at Radioactive Waste Management Limited (RWM) and I was asked by the editor if I would provide high-level comments on this Natural Hazards and Earth System Sciences journal paper (from here on 'The Paper').

My experience relevant to the review is twofold:

a) The requirement for RWM to calculate potential risks to future populations from buried radioactive waste over geological timescales requires quantification and treatment of many disparate uncertainties, and is in many ways analogous to the hazard risk assessments considered in The Paper.

b) I have recently written a report explaining in detail RWM's strategy for handling uncertainty [1], published earlier this year, so it is interesting to compare notes.

The Paper covers the issues that I would expect to see covered, from the need to build a model of the system, including consequences, and carry out a probabilistic analysis of some sort. The need to rely fairly extensively on expert judgement and the importance of effective elicitation methods is also clear. The Paper also recognises the challenges in communicating issues related to risk, assessment of risk, and uncertainty – although perhaps the section recognising the importance of good visualisation methods, could use some, ahem, pictures!

I would like to focus my comments on four aspects of The Paper in particular, which I will state up front because they interrelate to some extent:

1. The extent to which it is sensible to distinguish epistemic and aleatory uncertainty.

2. Probabilistic vs non-probabilistic methods for quantifying uncertainty.

3. The role of expert elicitation and methodology used.

4. The extent to which it is necessary to make assumptions.

**1. Epistemic and aleatory uncertainties.**

At the point of developing a model for assessing risks, for practical purposes, I think the distinction between aleatory uncertainty and epistemic uncertainty is moot. Such a model needs to account for all sources of uncertainty on the same footing, irrespective of type. When considering the results from a model, e.g. which parameters the performance measure is sensitive to, I accept that it may then become important which uncertainties are epistemic, and therefore potentially reducible.

The Paper says that epistemic uncertainties can be inherently difficult to represent as probabilities – I would say all uncertainties can be represented as probability distributions, the difficulty is more that people struggle to do this (hence the need for elicitation methods) and that on some occasions the probability distribution (if unbiased) is so wide that it is describing virtually no knowledge, so is therefore unhelpful.

**2. Probabilistic and non-probabilistic methods**

I found S2.5 of The Paper rather difficult to follow. It introduces the spectre of non-probabilistic methods for quantifying uncertainty, and talks rather too much about assumptions (see 4 below). I think a paper on good practice could offer more on the relative merits of different methods, but I detect some fence-sitting in this discussion as it unfolds. I am not an expert in non-probabilistic methods, but have a deep mistrust of them for the following reason.

I presume the reason they were devised is largely because people have great difficulty estimating probabilities without bias. But probability is mathematically the correct way to represent uncertainty. If any alternative model is used, then this must add additional significant model uncertainty to the overall system. It seems to me therefore that for such an alternative model to prove useful, people would need to be better at estimating the

required quantities for that model (whatever they are), than they are at estimating probabilities, by more than this additional model uncertainty. I am sceptical that this is ever true, therefore I am a strong advocate of a probabilistic approach, particularly because I believe that the difficulty in estimating probabilities without bias can be overcome by effective elicitation methods.

**3. Expert elicitation**

Good expert elicitation methods are essential when expert judgement is the only way to quantify uncertainty in model parameters. The Paper could be clearer whether elicitation is the process of quantifying uncertainty in a parameter in the form of a probability density function (PDF) by experts only, or includes eliciting other information from experts to e.g. what assumptions to make, conceptual models etc. It's the former that I am interested in commenting on here. (I've avoided the term elicitation in [1] and referred to the process throughout for clarity as 'uncertainty quantification by expert judgement'.)

The Cooke methodology cited in The Paper is an excellent method for combining PDFs from different experts taking into account both their knowledge and their skill in quantification of uncertainty, but a concern is that few experts actually contribute to the combined PDF because most are very poor at the latter, which isn't great use of resources. It is probably the best method for mathematically combining input from multiple experts, but unless significant steps are taken to minimise experts' bias, that will be the best of a bad bunch – maybe that's why on P5 The Paper recommends using as many experts as possible. Training and good facilitation can help here and I strongly support the comment on P6 of The Paper about the development of training methods for experts and facilitators alike – this is necessary.

I think an effective elicitation process should consist of three things:

1. People use quick mental heuristics to make decisions under uncertainty, which are subject to significant cognitive biases, usually resulting in over-confidence. There is therefore a need for an initial training exercise which highlights these natural biases. I have developed one using weather statistics which works effectively (see S6.2 of [1]).

2. The process of an expert generating a PDF needs to follow a structured approach designed to avoid these cognitive biases, especially 'anchoring' on an initial guess and failing to adjust enough for uncertainty. This is essentially done by asking the question in an indirect way ensuring that the expert considers extreme values (of the PDF) first (see S6.3 and S6.4 of [1]). Just asking experts to give ranges or properties of the distribution directly will not avoid bias.

3. If a group of experts are used, their inputs need to be combined. This can be done mathematically e.g. by the Cooke method, or the PDF can be generated by the group together by discussion aiming for consensus. There are pros and cons of both; we have tended to do the latter (see S6.4 of [1]).

I think The Paper would benefit from a paragraph on cognitive biases (see S5.1 of [1]) as the reason for requiring a formal elicitation methodology at the beginning of S2.3.

**4. Assumptions**

The Paper uses the word 'assumptions' far too often, in my view, and I don't think phrases like 'assumptions about the uncertainty' (e.g. P9) actually make sense – an assumption is something you make to avoid quantifying uncertainty, it has no place in the process of doing so. If authors really believe they are making assumptions about uncertainty instead of quantifying it, that doesn't leave us in a good place.

Clearly, there is a need for some assumptions in defining scenarios, and building a conceptual model for a given scenario, but care needs to be taken mixing assumptions with a probabilistic approach for two reasons. Assumptions exclude uncertainty, meaning the

results of any probabilistic analysis on the remaining uncertainty are conditional on the assumptions being correct, if the majority of uncertainty is excluded by assumptions, the probabilistic analysis may not prove very helpful.  If very conservative assumptions are used for a subset of parameters in a model as an alternative to quantifying uncertainties, this can cause an otherwise probabilistic assessment to be biased, which again may reduce its usefulness (see S4.3 of [1]).

P3 of The Paper cautions about making assumptions about the form of probability distributions, and is right to highlight this as an issue.  In my view when using expert judgement to quantify uncertainty in a parameter, it is best to not to constrain the shape of the distribution.  Therefore, I would recommend using a general cumulative distribution – that is a PDF where the cumulative distribution function (CDF) is piecewise linear – which allows the expert to put as much or as little structure on the PDF (particularly in the tails) as he or she is able to.  Another important issue is the correct choice of scale, linear or logarithmic.  These issues are discussed further in S2.3, S2.4, S6.3 and S6.4 of [1].

I hope these comments are useful and am happy to discuss them if that would be helpful.

Mike ([mike.poole@nda.gov.uk](mailto:mike.poole@nda.gov.uk))

[1]      RWM, *Geological Disposal: Methods for Management and Quantification of Uncertainty*, NDA/RWM/153, 2017.  This report is available here: https://rwm.nda.gov.uk/publication/methods-for-management-and-quantification-of-uncertainty/

---

## Referee Comment (RC2) · Anonymous Referee #2 · 12 Oct 2017

The paper has considerably changed - and improved - compared to an earlier version. Still I would suggest to consider some major issues:

1. I struggle a bit to imagine a potential audience for whom the paper is really useful (i.e., introducing something new rather than summarising known issues). Since the paper aims to lay out what constitutes good practice, a natural target audience would be not only experts in the field, but also researchers trying to get into the field. But several parts, in particular section two, is extremely dense and difficult to read for non-experts. One example is the explanation of micro-correlations, which is essentially not understandable from the text. Here it would help to provide more (or better) expla-

nations and, for the different analysis approaches, a comparative discussion to make the paper more accessible. During rewriting, the authors should also consider the terminology. As stated previously, the term "simulator" is widely used in the statistics community, but a climate modeller, e.g., might not realise that it might refer to a climate model. Such key terms should simply be defined when used for the first time.

2. I still find the manuscript somewhat unbalanced. Almost one third of the references are self citations. In particular I wonder whether well known general statements have to be backed up by self citations if there might be older standard works available (e.g., page 4, line 31).

3. It should be worked out that the distinction between aleatory and epistemic uncertainty is often not clear cut, but may evolve in time, or may be simply due to pragmatic reasons. So often we treat uncertainties as aleatory rather than assuming that they are intrinsically stochastic. This is implicitly mentioned in the manuscript, but could be made more explicit.

4. I find the recommendations regarding the treatment of future projections (last paragraph on page 10) somewhat misleading. It is not in general justified to treat GCM or downscaled simulations equally - this is a question of what the models are supposed to represent. A GCM might miss local feedbacks and thus provide an implausible change signal. So here a decision has to be drawn on a case-by-case basis whether GCMs should be considered or downscaling is necessary (we have three possibilities: 1. GCM and RCM can be treated equally, 2. RCMs are credible, GCMs implausible, 3. the RCM is doing something wrong and the GCM is credible (of course one might also consider a fourth case where no model is fit for purpose.)

5. Related to this issue: the discussion of RCMs (page 16, line 16-19) should be modified. The cited studies suggest that standard RCMs (which do not resolve convection) do not realistically simulate the response of extreme convective precipitation, but very high resolution RCMs may indeed (which explicitly represent deep convection). This

finding also backs up my statement 4 (that different types of models serve different purposes and should not be treated equally).

Final comment: I mildly disagree with the other reviewer regarding non-probabilistic approaches. In particular wrt time varying risk and extrapolation (such as in the case of climate change), the issue is not about estimating probabilities. Here a Bayesian approach might help to formally attach probabilities to certain future simulations, but I am wondering whether this would not give false confidence - because in many situations we simply do not have the knowledge to come up with meaningful prior distributions.

---

## Author Comment (AC1) · 28 Nov 2017

see attached file.

Please also note the supplement to this comment:
https://www.nat-hazards-earth-syst-sci-discuss.net/nhess-2017-251/nhess-2017-251-AC1-supplement.pdf
* * *

---

## Author Comment (AC2) · 28 Nov 2017

**nhess-2017-251**
**Epistemic uncertainties and natural hazard risk assessment. 2. What should constitute good practice?**

**Response to Referee's Comments.**

We had been hoping that the discussion to Part 1 of this paper would have also been closed before responding to the comments below, but it seems that only one referee comment has, as yet, been received on Part 1.

**Referee 1**
My name is Mike Poole and I am a Senior Research Manager at Radioactive Waste Management Limited (RWM) and I was asked by the editor if I would provide high-level comments on this Natural Hazards and Earth System Sciences journal paper (from here on 'The Paper').

My experience relevant to the review is twofold:

a) The requirement for RWM to calculate potential risks to future populations from buried radioactive waste over geological timescales requires quantification and treatment of many disparate uncertainties, and is in many ways analogous to the hazard risk assessments considered in The Paper.

b) I have recently written a report explaining in detail RWM's strategy for handling uncertainty [1], published earlier this year, so it is interesting to compare notes.

The Paper covers the issues that I would expect to see covered, from the need to build a model of the system, including consequences, and carry out a probabilistic analysis of some sort. The need to rely fairly extensively on expert judgement and the importance of effective elicitation methods is also clear. The Paper also recognises the challenges in communicating issues related to risk, assessment of risk, and uncertainty – although perhaps the section recognising the importance of good visualisation methods, could use some, ahem, pictures!

Part 1 of the paper does have more visualisations, but we can add more examples here.

I would like to focus my comments on four aspects of The Paper in particular, which I will state up front because they interrelate to some extent:

1. The extent to which it is sensible to distinguish epistemic and aleatory uncertainty.

2. Probabilistic vs non-probabilistic methods for quantifying uncertainty.

3. The role of expert elicitation and methodology used.

4. The extent to which it is necessary to make assumptions.

1. Epistemic and aleatory uncertainties.
At the point of developing a model for assessing risks, for practical purposes, I think the distinction between aleatory uncertainty and epistemic uncertainty is moot. Such a model needs to account for all sources of uncertainty on the same footing, irrespective of type. When considering the results from a model, e.g. which parameters the performance measure is sensitive to, I accept that it may then become important which uncertainties are epistemic, and therefore potentially reducible.
The Paper says that epistemic uncertainties can be inherently difficult to represent as probabilities – I would say all uncertainties can be represented as probability distributions, the difficulty is more that people struggle to do this (hence the need for elicitation methods) and that on some occasions the probability distribution (if unbiased) is so wide that it is describing virtually no knowledge, so is therefore unhelpful.

I think we have to disagree with the referee here. This comment, and others below, is clearly conditioned on his statement below that probability is mathematically the correct way to represent uncertainty. In fact it can be argued that all uncertainties are epistemic and reducible, but we will sometimes choose to represent them as if they were aleatory using probabilities. But even then any uncertainty estimation is going to be conditional on the assumptions that are made, even if that is only the choice of distribution that is used to represent the potential outcomes (unbounded normal or bounded beta for example – both may fit the evidence available well). We would therefore argue that the recognition of sources of uncertainty as epistemic is an important element in good practice to focus attention on what the appropriate assumptions might be. We would also note that even if the resulting uncertainties are wide, they are not necessary unhelpful (see Almeida et al., NHESS, 2017, for example).

2. Probabilistic and non-probabilistic methods
I found S2.5 of The Paper rather difficult to follow. It introduces the spectre of nonprobabilistic methods for quantifying uncertainty, and talks rather too much about assumptions (see 4 below). I think a paper on good practice could offer more on the relative merits of different methods, but I detect some fence-sitting in this discussion as it unfolds. I am not an expert in non-probabilistic methods, but have a deep mistrust of them for the following reason.

I presume the reason they were devised is largely because people have great difficulty estimating probabilities without bias. But probability is mathematically the correct way to represent uncertainty. If any alternative model is used, then this must add additional significant model uncertainty to the overall system. It seems to me therefore that for such an alternative model to prove useful, people would need to be better at estimating the 2 required quantities for that model (whatever they are), than they are at estimating probabilities, by more than this additional model uncertainty. I am sceptical that this is ever true, therefore I am a strong advocate of a probabilistic approach, particularly because I believe that the difficulty in estimating probabilities without bias can be overcome by effective elicitation methods.

Again, I think we have to disagree here (as does, diplomatically, Referee 2). We can envisage applications where the use of bounded fuzzy possibilities would lead to more constrained and realistic uncertainty estimates than unbounded probabilities (that can in the worse cases give probability to negative values of variables that can only be positive). To suggest that estimating probabilities without bias can be overcome by effective elicitation methods is, to say the least, surprising when there are many cases of experts showing collective bias. This perhaps comes from this being the method that is accepted as good practice in nuclear regulation. Experience suggests it might not be so easy to get unbiased elicitation in natural hazards.

3. Expert elicitation

Good expert elicitation methods are essential when expert judgement is the only way to quantify uncertainty in model parameters. The Paper could be clearer whether elicitation is the process of quantifying uncertainty in a parameter in the form of a probability density function (PDF) by experts only, or includes eliciting other information from experts to e.g. what assumptions to make, conceptual models etc. It's the former that I am interested in commenting on here. (I've avoided the term elicitation in [1] and referred to the process throughout for clarity as 'uncertainty quantification by expert judgement'.)

The Cooke methodology cited in The Paper is an excellent method for combining PDFs from different experts taking into account both their knowledge and their skill in quantification of uncertainty, but a concern is that few experts actually contribute to the combined PDF because most are very poor at the latter, which isn't great use of resources.

This is a slight misrepresentation of the situation as far as the Classical Model is concerned: all participating experts contribute to the combined PDF (or other elicited quantity), to the extent to which they add information and statistical accuracy to the combination. Before (Classical Model) calibration, abundant experience indicates that there is no way of knowing a priori who in a group will perform more strongly than others so, In this way, the available (expert) "resources" are first accessed and then used optimally.

It is probably the best method for mathematically combining input from multiple experts, but
unless significant steps are taken to minimise experts' bias, that will be the best of a bad bunch – maybe that's why on P5 The Paper recommends using as many experts as possible. Training and good facilitation can help here and I strongly support the comment on P6 of The Paper about the development of training methods for experts and facilitators alike – this is necessary.

I think an effective elicitation process should consist of three things:

1. People use quick mental heuristics to make decisions under uncertainty, which are subject to significant cognitive biases, usually resulting in over-confidence. There is therefore a need for an initial training exercise which highlights these natural biases. I have developed one using weather statistics which works effectively (see S6.2 of [1]).

2. The process of an expert generating a PDF needs to follow a structured approach designed to avoid these cognitive biases, especially 'anchoring' on an initial guess and failing to adjust enough for uncertainty. This is essentially done by asking the question in an indirect way ensuring that the expert considers extreme values (of the PDF) first (see S6.3 and S6.4 of [1]). Just asking experts to give ranges or properties of the distribution directly will not avoid bias.

3. If a group of experts are used, their inputs need to be combined. This can be done mathematically e.g. by the Cooke method, or the PDF can be generated by the group together by discussion aiming for consensus. There are pros and cons of both; we have tended to do the latter (see S6.4 of [1]).

I think The Paper would benefit from a paragraph on cognitive biases (see S5.1 of [1]) as the reason for requiring a formal elicitation methodology at the beginning of S2.3.

We can extend the discussion to consider this.

4. Assumptions
The Paper uses the word 'assumptions' far too often, in my view, and I don't think phrases like 'assumptions about the uncertainty' (e.g. P9) actually make sense – an assumption is something you make to avoid quantifying uncertainty, it has no place in the process of doing so. If authors really believe they are making assumptions about uncertainty instead of quantifying it, that doesn't leave us in a good place.

We find this reaction a bit difficult to understand given the conditional nature of any uncertainty estimate (see earlier comment).  The referee seems to have a more restricted view of the use of the word.  In our paper it is noted that choices and therefore assumptions are necessary about how to handle both aleatory and epistemic uncertainties in making judgements about sources of uncertainty and their intereactions.   Certainly one such choice is to define appropriate multi-variate probability distributions (as again is comment in nuclear regulation applications).   In natural hazards, recent work in the joint occurrences of hazards, takes just this approach.   But in many cases there is not enough observations to define properly the tail behaviours.   Thus the outcomes will depend on the choice of appropriate distributions (and copulas to represent covariance – for which there is a large choice) within the probabilistic framework.   These assumptions are actually examples of epistemic uncertainties – but we do not normally give probabilities to the possible different choices, we record what has been assumed (or it is already specified by a particular regulatory framework).   We will consider each use of the word assumption in the paper and see where it can be clarified.

Clearly, there is a need for some assumptions in defining scenarios, and building a conceptual model for a given scenario, but care needs to be taken mixing assumptions with a probabilistic approach for two reasons. Assumptions exclude uncertainty, meaning the 3 results of any probabilistic analysis on the remaining uncertainty are conditional on the assumptions being correct, if the majority of uncertainty is excluded by assumptions, the probabilistic analysis may not prove very helpful. If very conservative assumptions are used for a subset of parameters in a model as an alternative to quantifying uncertainties,

this can cause an otherwise probabilistic assessment to be biased, which again may reduce its usefulness (see S4.3 of [1]).

P3 of The Paper cautions about making assumptions about the form of probability distributions, and is right to highlight this as an issue. In my view when using expert judgement to quantify uncertainty in a parameter, it is best to not to constrain the shape of the distribution. Therefore, I would recommend using a general cumulative distribution – that is a PDF where the cumulative distribution function (CDF) is piecewise linear – which allows the expert to put as much or as little structure on the PDF (particularly in the tails) as he or she is able to. Another important issue is the correct choice of scale, linear or logarithmic. These issues are discussed further in S2.3, S2.4, S6.3 and S6.4 of [1].

We note that the use of a piecewise linear CDF is a choice (or assumption in our sense), and that other assumptions could be made (including the use of probability boxes to allow for uncertainty in the form of the CDF).   We note also that the choice of a "correct" scale is also an assumption in our sense.

I hope these comments are useful and am happy to discuss them if that would be helpful.

Mike (mike.poole@nda.gov.uk)

[1] RWM, Geological Disposal: Methods for Management and Quantification of Uncertainty, NDA/RWM/153, 2017. This report is available here: https://rwm.nda.gov.uk/publication/methods-for-management-and-quantification-ofuncertainty/

**Anonymous Referee #2**

The paper has considerably changed - and improved - compared to an earlier version. Still I would suggest to consider some major issues:

1. I struggle a bit to imagine a potential audience for whom the paper is really useful (i.e., introducing something new rather than summarising known issues). Since the paper aims to lay out what constitutes good practice, a natural target audience would be not only experts in the field, but also researchers trying to get into the field. But several parts, in particular section two, is extremely dense and difficult to read for non- experts. One example is the explanation of micro-correlations, which is essentially not understandable from the text. Here it would help to provide more (or better) explanations and, for the different analysis approaches, a comparative discussion to make the paper more accessible.

Our aim, more evident perhaps when Paper 1 of the series is added, was to reach a cross-disciplinary audience and raise awareness of issues associated with epistemic uncertainties across natural hazards given the different practice in different hazard areas. Clearly, this means it is going to be difficult to be accessible to all in such a review – but sufficient references are given such that readers can follow-up points if necessary.

2. During rewriting, the authors should also consider the terminology. As stated previously, the term "simulator" is widely used in the statistics community, but a climate modeller, e.g., might not realise that it might refer to a climate model. Such key terms should simply be defined when used for the first time.

We can avoid the word simulator and will check other technical terms.

2. I still find the manuscript somewhat unbalanced. Almost one third of the references are self citations. In particular I wonder whether well known general statements have to be backed up by self citations if there might be older standard works available (e.g., page 4, line 31).

But we are writing from our own experience and opinions of authors who are pushing the frontiers of uncertainty estimation, primarily backed up by our own applications (so one third might be considered to be quite low). We are arguing for what we think it is important to consider as good practice from that experience.

3. It should be worked out that the distinction between aleatory and epistemic uncertainty is often not clear cut, but may evolve in time, or may be simply due to pragmatic reasons. So often we treat uncertainties as aleatory rather than assuming that they are intrinsically stochastic. This is implicitly mentioned in the manuscript, but could be made more explicit.

This is a good example of how different people understand these words in different ways (stochastic in the form of stochastic models normally assumes aleatory variations). We had hoped that this differentiation was clear, but will reconsider the text accordingly.

3.  I find the recommendations regarding the treatment of future projections (last paragraph on page 10) somewhat misleading. It is not in general justified to treat GCM or downscaled simulations equally - this is a question of what the models are supposed to represent. A GCM might miss local feedbacks and thus provide an implausible change signal. So here a decision has to be drawn on a case-by-case basis whether GCMs should be considered or downscaling is necessary (we have three possibilities: 1. GCM and RCM can be treated equally, 2. RCMs are credible, GCMs implausible, 3. the RCM is doing something wrong and the GCM is credible (of course one might also consider a fourth case where no model is fit for purpose.)

Well, we only said that we CAN give them equal weight and other choices are certainly possible (we can cite a case where different weights have been given) but it is certainly a rather common choice (including in IPCC presentations).

4.  Related to this issue: the discussion of RCMs (page 16, line 16-19) should be modified. The cited studies suggest that standard RCMs (which do not resolve convection) do not realistically simulate the response of extreme convective precipitation, but very high resolution RCMs may indeed (which explicitly represent deep convection). This finding also backs up my statement 4 (that different types of models serve different purposes and should not be treated equally).

Mmmm .. .. .. but RCMs are also nested within GCM provided boundary conditions so is this not more a question of belief in the process representations as being better rather than a real capability of being able to predict extremes?   We will, however, modify the text with appropriate references.

Final comment: I mildly disagree with the other reviewer regarding non-probabilistic approaches. In particular wrt time varying risk and extrapolation (such as in the case of climate change), the issue is not about estimating probabilities. Here a Bayesian approach might help to formally attach probabilities to certain future simulations, but I am wondering whether this would not give false confidence - because in many situations we simply do not have the knowledge to come up with meaningful prior distributions.

We can but agree.